# Nonlinear Mixed-Effect Pharmacokinetic Modeling and Distribution of Doxycycline in Healthy Female Donkeys after Multiple Intragastric Dosing–Preliminary Investigation

**DOI:** 10.3390/ani11072047

**Published:** 2021-07-09

**Authors:** Ronan J. J. Chapuis, Joe S. Smith, Hilari M. French, Felix Ngosa Toka, Erik W. Peterson, Erika L. Little

**Affiliations:** 1Department of Clinical Sciences, Ross University School of Veterinary Medicine, Basseterre, Saint Kitts and Nevis; rchapuis@rossvet.edu.kn (R.J.J.C.); hfrench@rossvet.edu.kn (H.M.F.); erpeterson@rossvet.edu.kn (E.W.P.); 2Large Animal Clinical Sciences, College of Veterinary Medicine, University of Tennessee, Knoxville, TN 37996, USA; animal197@gmail.com; 3Department of Biomedical Sciences, Ross University School of Veterinary Medicine, Basseterre, Saint Kitts and Nevis; ftoka@rossvet.edu.kn

**Keywords:** doxycycline hyclate, intragastric, donkey, pharmacokinetics, serum, synovial fluid, urine, endometrium

## Abstract

**Simple Summary:**

Rational use of antibiotic is of most importance for human and animal health. Donkeys have differences in metabolism compared to horses, and drug dosage should not be extrapolated from horses’ data. Doxycycline, a common antibiotic used in equine medicine, has never been investigated in donkeys. The aim of this preliminary study was to describe the concentrations of doxycycline obtain in serum and other body tissue and fluid following its oral administration in donkeys at the recommended horse dosage. Doxycycline was administered to eight healthy, adult jennies. Serum, urine, synovial fluid and uterine tissue were collected. Doxycycline concentrations were measured with a commercial ELISA. A pharmacological model was used to analyze the serum concentration and calculate some pharmacological parameters. Results suggest that doxycycline is well absorbed following oral administration and calculated serum parameters suggest high tissue distribution. However, the concentration of doxycycline reached in all fluids and tissues analyzed would unlikely result in therapeutic concentration against common equine pathogens. Further investigations are warranted. This data can be used for designing future studies of doxycycline in donkeys. In the meantime, oral doxycycline at the horse dosage should not be consider a suitable treatment in donkeys until proven efficacious.

**Abstract:**

Doxycycline (DXC) is a broad-spectrum antibacterial antimicrobial administered to horses for the treatment of bacterial infections which may also affect donkeys. Donkeys have a different metabolism than horses, leading to differences in the pharmacokinetics of drugs compared to horses. This study aimed to describe the population pharmacokinetics of DXC in donkeys. Five doses of DXC hyclate (10 mg/kg) were administered via a nasogastric tube, q12 h, to eight non-fasted, healthy, adult jennies. Serum, urine, synovial fluid and endometrium were collected for 72 h following the first administration. Doxycycline concentration was measured by competitive enzyme immunoassay. Serum concentrations versus time data were fitted simultaneously using the stochastic approximation expectation-maximization algorithm for nonlinear mixed effects. A one-compartment model with linear elimination and first-order absorption after intragastric administration, best described the available pharmacokinetic data. Final parameter estimates indicate that DXC has a high volume of distribution (108 L/kg) as well as high absorption (10.3 h^−1^) in donkeys. However, results suggest that oral DXC at 10 mg/kg q12 h in donkeys would not result in a therapeutic concentration in serum, urine, synovial fluid or endometrium by comparison to the minimum inhibitory concentration of common equine pathogens. Further studies are recommended to identify appropriate dosage and dosing intervals of oral DXC in donkeys.

## 1. Introduction

Donkeys (*Equus asinus*) remain a source of income in the cosmetic, tourism, farming, and food (milk and meat) industries. Even if the worldwide donkey population is decreasing, donkeys have taken a greater place in society as companion animals, and now are used for emotional support (asino- or onotherapy), increasing the demand for veterinary medical management [1]. In recent years, donkey farming has gained popularity in several countries with a growing interest in milk production [2]. Indeed, consumption of donkeys’ milk is encouraged because of its beneficial human health properties. When compared to cow’s milk, donkeys’ milk is a better milk replacement option for infants and a suitable milk source for people suffering from cow’s milk allergies [2,3]. It is important to conduct pharmacological studies to investigate medical treatment options for donkeys. Furthering research in this area will help maintain the welfare of the reared donkeys, as well as guaranteeing human biosecurity, specifically regarding withdrawal time to apply in food production. Donkeys are classified as a minor species by the Food and Drug Administration and no drugs are labeled for use in this species. Therefore, drugs, including antibacterial antimicrobials, are used in an extra-label manner in donkeys. However, the metabolic rate and cellular water content are higher in donkeys than in horses. Most antibacterials have a higher clearance, lower mean residential time and shorter half-life in mules and donkeys. Higher dosages and/or shorter intervals between dosage are often required in this species when compared to horses; therefore, dosages should not be directly extrapolated from horses’ pharmacologic data [4]. These drug metabolism discrepancies present an obvious need for pharmacologic studies in *Equus asinus* species to provide evidence-based medical treatments to these animals, as well as for establishing withdrawal times.

Pharmacologic studies in donkeys of antibacterials include ß-lactams (amoxicillin, ampicillin, penicillin G), aminoglycosides (amikacin, gentamicin), fluoroquinolones (danofloxacin, enrofloxacin, marbofloxacin, norfloxacin), oxytetracycline, sulfonamides (sulfadimidine, sulfadiazine, sulfadimethoxine, sulfamethoxazole, sulfamethoxypyridazine), and trimethoprim [4]. However, even if oral medication is desirable in equine medicine for ease, safety and compliance, the sole oral antibacterial studied in donkeys to date is norfloxacin. Norfloxacin has demonstrated poor bioavailability and is thus suggested not usable based on the minimum inhibitory concentration (MIC) reported of common susceptible bacteria in equine [5]. Furthermore, following antimicrobial stewardship principles, pharmacologic studies in veterinary medicine should focus priorities on lower-ranked critical molecules for human medicine [6], therefore doxycycline warrants some investigation.

Doxycycline (DXC) is a semi-synthetic analog of tetracycline and has broad-spectrum co-dependent bacteriostatic properties [7]. The pharmacokinetics of DXC following oral administration in adult healthy horses have been studied [8,9,10,11,12,13,14]. These studies suggested that oral DXC at 10 mg/kg q12 h reached adequate concentration for the treatment of susceptible bacteria with a MIC lower than 0.25 µg/mL in serum, peritoneal fluid, synovial fluid, endometrium, pulmonary epithelial lining fluid, and urine. If administered orally at 20 mg/kg q12 h, the concentration of DXC reached a concentration of 29 µg/mL in polymorphonuclear leukocytes [10]. However, multiple horses experienced colic, and one horse died from fatal colitis. Other side effects reported in horses following oral administration of DXC include anorexia and photosensitization [10,12].

Some authors suggested the clinical efficacy of oral DXC in horses for the treatment of intra-abdominal or cutaneous abscesses from Glanders [15,16], neuroborreliosis [17], granulocytic ehrlichiosis, neorickettsiosis, and leptospirosis [10]. Natural infection of neorickettsiosis (Potomac horse fever) has not been reported in the donkey but could be replicated experimentally [18]. Antibodies against *Anaplasma phagocytophilum* have been detected in donkeys [19]. There are no reports of intra-abdominal abscessation in donkeys, but pathogens that can cause abscessation such as *Streptococcus zooepidemicus* and *S. equi* have been isolated in donkeys [20]. Glanders, neuroborreliosis, leptospirosis have also been reported in donkeys [21]. Therefore, DXC warrants investigation in donkeys to support its use for the medical management of these diseases in this species.

This study aimed to investigate the pharmacokinetics of DXC by determining its distribution in serum, urine, synovial fluid, and endometrium after intragastric administrations of 5 doses of DXC at 10 mg/kg PO, q12 h in healthy jennies.

## 2. Materials and Methods

The study was approved by the Institutional Animal Care and Use Committee at Ross University School of Veterinary Medicine (RUSVM), number 16-10-032.

### 2.1. Animals

Eight healthy adult Caribbean crossbred jennies, from 4 to 12 years old, weighing between 120 kg and 180 kg (weights rounded to the nearest 10 kg), were randomly selected from the teaching herd of the RUSVM. Inclusion criteria included normal physical examination and complete blood counts. All jennies were housed in an outside dry pen and were fed ad libitum guinea grass and water.

Jennies were monitored during the study until the last sample collection (72 h after initiation of the medication) and then regularly as part of the teaching herd. All jennies remained healthy 4 years after the study.

### 2.2. Experimental Design and Sample Collection

Doxycycline hyclate 100 mg coated tablets (West-Ward Pharmaceuticals Corp., Eatontown, NJ, USA) were dissolved in 20 mL of tap water to prepare individual doses of 10 mg/kg. The DXC was administered via a nasogastric tube, which was flushed with 500 mL of tap water prior to removal to ensure complete delivery of the DXC. Each jenny received 10 mg/kg of the drug at time point 0 and then q12 h for five doses (0, 12, 24, 36, and 48 h). Jennies were sedated with xylazine hydrochloride (0.7 mg/kg) intravenously as needed for sample collection.

Blood samples were collected with a 16 G polyurethane intravenous catheter placed in the jugular vein prior to the first DXC administration (time 0), and 15, 30 min, as well as 1, 2, 6, 12, 18, 24, 24.5, 28, 30, 36, 36.5, 42, 48, 48.5, 54, 60, and 72 h following the first administration. The samples were placed in plain glass blood collection tubes (Covidien, Mansfield, MA, USA), tubes were centrifuged to separate serum (3000 rpm for 15 min), and serum was stored at −80 °C until analyzed.

Urine samples were collected aseptically using a stallion urinary catheter prior to DXC administration (time 0) and 24, 36, 48, and 60 h following the first administration. The samples were placed in Eppendorf tubes and stored at −80 °C until analyzed.

One mL of synovial fluid sample was collected from the tibiotarsal joints aseptically by arthrocentesis, alternating between right and left tarsus. Samples were collected prior to the DXC administration (time 0), and 24, 36, 48, and 60 h following the first administration. The samples were placed in glass blood collection tubes, centrifuged (3000 rpm for 15 min) and supernatant placed in Eppendorf tubes and stored at −80 °C until analyzed.

Endometrial tissue samples were collected by obtaining uterine biopsies using endometrial biopsy forceps. Biopsies were collected 51, 60 and 72 h following the first DXC administration. The samples were placed in sterile, isotonic saline-filled Eppendorf tubes and stored at −80 °C until analyzed.

All the samples were collected prior to DXC administration when the time of sampling and administration of DXC were concomitant.

### 2.3. Measure of DXC

Total doxycycline was measured by competitive enzyme immunoassay (MaxSignal ^®^ Doxycycline ELISA Test Kit, #1083, BIOO Scientific Corp, Austin, TX, USA). The manufacturer reports the sensitivity of DXC detection in meat/meat products/fish/shrimp/butter and the specificity of the kit at 1.5 ng/g and 100.0%, respectively. The kit is manufactured in accordance with the international quality standard ISO 9001:2008. This ELISA kit was used in previous studies for measuring the release of DXC from nanotube surface-treated dental implants loaded with DXC [22], and regulatory residue surveillance [23].

The extraction protocol was adapted from the protocol of DXC concentration analysis in animal tissues as described in the manufacturer manual for meat and meat products (#1083-01D, V.14.06, 2014). For measuring DXC in serum, urine and synovial fluid, the following adaptations were performed: 100 µL of the sample was used instead of 1 g, and the first mixed sample was only shaken and not vortexed.

The endometrium biopsies were first disrupted in a sterile tissue homogenizer to a fine pulp. One milliliter of phosphate-buffered saline was added, and the homogenate was mixed and vortexed at high speed for 2–3 min. The homogenate was placed on ice for 5 min and then clarified at 1000 rpm for 5 min with the centrifuge set at 4 °C. The resulting supernatant was subjected to another round of centrifugation. One hundred microliters of the supernatant were used and processed for DXC analysis as described above for the fluid samples.

For serum samples, the intra- and interassay coefficients of variation were 18.1% and 13.1%, respectively. For urine samples, the intra- and interassay coefficients of variation were 19.5% and 15.8%, respectively. For synovial fluid samples, the intra- and interassay coefficients of variation were 12.4% and 5.2%, respectively. For uterine biopsy samples, the intra- and interassay coefficients of variation were 8.9% and 8.2%, respectively.

### 2.4. Pharmacokinetic Analysis in Serum

#### 2.4.1. Non-Linear Mixed Effect Model Building and Evaluation

All serum concentration DXC data from all donkeys were pooled for nonlinear mixed-effect (NLME) analysis. Doxycycline serum concentration vs. time courses was evaluated by the stochastic approximation expectation maximization (SAEM) algorithm implemented in the Monolix Suite 2019R2 (Lixoft). Individual model parameters were determined post hoc using the full posterior distribution of the mean as previously described [24,25].

The model was evaluated as described for extravascular administration by Wang et al. [24,25]. Convergence of the SAEM algorithm was evaluated by inspecting the stability of the fixed and random effect parameters and estimating the log-likelihood after the exploratory period (after 1000 iterations of SAEM). Standard goodness-of-fit diagnostics, including individual predictions versus observations, the distribution of weighed residuals (IWRES), and normalized prediction distribution centers (NPDE) were used to assess the performance of the candidate models. Prediction distributions from 500 Monte Carlo simulations were used to evaluate the ability of the final model to reproduce the variability in the observed pharmacokinetic data. Bayesian information criteria (BIC) were used to select between competing structural models. BIC was selected over the Akaike Information Criterion as it favors data that tends to be parsimonious [24,25,26].

Additionally, an estimate of apparent systemic clearance (CL/F) was calculated using:K_el_ = CL/V_d_
where K_el_ is the linear elimination rate; V represents the apparent volume of distribution during the terminal phase after non-intravenous administration (V_z_/F), and CL (CL/F) representing the clearance for extravascular drug administration.

#### 2.4.2. Estimation of Parameter Correlation

Parameter correlation was estimated as previously described [24,25]. Plots were visually inspected, and results from the Pearson’s correlation tests were used to inform our choice of correlations between model parameters. In agreement with previous literature [27,28], multiple samples from the last SAEM (posterior distribution) obtained at the last iteration of the SAEM were used during the evaluation of parameter correlations. The final inclusion of correlations in the structural model was determined by the precision of the parameter estimates.

## 3. Results

Eleven of the 160 analyzed serum samples, 2 of the 40 analyzed urine samples, and 1 of the 40 synovial fluid samples were eliminated due to sampling errors. None of the jennies exhibited side effects.

### 3.1. Serum

#### 3.1.1. Pharmacokinetic Model Evaluation

A one-compartment model with linear elimination and first-order absorption after intragastric administration, best described the available pharmacokinetic data based on standard goodness of fit plots (Figure 1), as well as BIC (Figure 2 and Figure 3).

The predictive performances and robustness of fit of the final model were supported by the goodness of fit plot inspection, displayed on a log-scale (Figure 1) to better evaluate the quality of fit as described by Nguyen et al. [29]. In this model of absorption, the 1st order absorption rate was represented by Ka.

#### 3.1.2. Estimation of Parameters and Model Evaluation

Table 1 and Table 2 provides final estimates of parameters. The precision of the final estimates was high (relative standard error (RSE) ≤ 15%) for K_el_, satisfactory for V, (RSE ≤ 25%) and lower for K_a_ (RSE ≤ 50%) (Appendix A). This model could reproduce the variability amongst individuals with little individual error (Figure 2 and Figure 3). Of note is the presence of inter-individual variability of one jenny (#1, Figure 2 and Figure 3) compared to the other subjects. The total clearance (CL/F) was estimated to be 2.73 L/kg/hr.

The mean peak concentration of DXC (C_max_) measured in the serum was 0.19 ± 0.13 μg/mL (median 0.13 [0.09–0.26] μg/mL) and 0.21 ± 0.21 μg/mL (median 0.11 [0.09–0.24] μg/mL), following the first and the last doses respectively (Figure 4). The maximum mean DXC concentration in the serum was 0.27 ± 0.25 μg/mL (median 0.13 [0.09–0.49] μg/mL), measured at 18 h, between the second and third doses.

### 3.2. Urine

Figure 5 depicts the individual concentration of DXC in urine. The maximum mean DXC concentration in the urine was 0.13 ± 0.02 μg/mL (median 0.13 [0.12–0.14] μg/mL), measured at 48 h. The mean concentration in urine was noted to be lower than in serum for all measurement time points (Figure 4).

### 3.3. Synovial Fluid

Figure 4 and Figure 6 depict the individual concentration of DXC in synovial fluid. The maximum mean DXC concentration in the synovial fluid was 0.06 ± 0.03 μg/mL (median 0.07 [0.02–0.09] μg/mL), measured at 48 h. The mean concentration of DXC in synovial fluid was noted to be lower than in serum for all measurement time points (Figure 4).

### 3.4. Endometrium

Figure 4 and Figure 7 depict the total DXC measured in endometrium biopsy.

## 4. Discussion

In the present preliminary study, serum pharmacokinetic data from repeated intragastric dosing of DXC at 10 mg/kg q12 h in donkeys was reported using the NLME modeling. Additionally, concentrations of DXC in synovial fluid, urine, and uterine tissue were reported. To the authors’ knowledge, this is the first study describing oral DXC pharmacokinetics in donkeys.

Due to the sparse numbers of measurements of serum DXC concentration, the NLME model approach to estimate pharmacokinetic parameters is recommended [30]. To use a statistical moment (i.e., non-compartmental) approach, future extravascular pharmacokinetic studies should include more intense sampling, especially during the absorption phase, prior to reaching maximal concentration, as the estimated absorption can be influenced by the sample collection schedule.

The data was best described by a one-compartment model with linear elimination after first-order absorption from repeated oral administration. Parameters from the final model suggest a high volume of distribution (V_z_/F, 108 L/kg) and absorption (K_a_, 10.3 h^−1^) of DXC in donkeys, but a low elimination rate (K_el_, 0.0253 h^−1^). There is a paucity of information about the volume of distribution (V_d_) of DXC in horses [31]. However, DXC is an extremely lipophilic molecule, and partitioning in the extravascular space should result in high V_d_. The high V_z_/F calculated in the present study can explain the low serum concentration. However, as V_z_/F is calculated from serum concentration, it is also possible that V_z_/F was artificially elevated from the serum values and the V_d_ would be lower. Other parameters, such as high clearance, can cause low serum concentration of DXC, and V_z_/F can be different from the real V_d_. Clearance of DXC was estimated at 2.73 L/kg/h in the current study, whereas the previous study reported a lower value of 42.5 mL/kg/h (0.0425 L/kg/h) in healthy horses [9]. The elimination rate (K_el_) in the present study was estimated from the slope in the elimination phase. Estimated K_el_ is lower (0.0253 h^−1^) than the ones reported in healthy horses ranging from 0.05 ± 0.04 to 0.2 ± 0.0 h^−1^ [10,11,12,14]. This suggests that DXC has a longer half-life in donkeys than in horses. However, the finding of a higher clearance contradicts the findings of longer half-life and lower elimination rate compared to horses and would require further investigations. Previous studies reported higher clearance, lower mean residential time and shorter half-life of most antibacterials in donkeys compared to horses; however, because exceptions exist, the pharmacokinetics of each antibacterial should be investigated in donkeys prior to reporting conclusions [4,32].

Urine DXC concentration seems lower in donkeys than in horses following the same dosage regimen [9]. Therefore, renal excretion of DXC might be lower in donkeys than in horses. Fecal excretion following hepatic metabolism is the main route of DXC excretion [9] and should be studied in donkeys. As donkeys have higher metabolic rates than horses [4], the low serum concentration of DXC and high clearance might be due to a higher hepatic metabolism in donkeys compared to horses. Nevertheless, future studies should investigate the concentration of DXC in other tissues to confirm a high V_d_, investigate the route and the metabolic rate of elimination of DXC, and be prepared for a high V_d_ of DXC in donkeys to design the sampling methods accordingly. Half-life is a composite of clearance and V_d_. As such, the half-life appears longer in the current study; and as the clearance is higher than in horses, it can be hypothesized that the longer half-life is related to a larger V_d_ in donkeys. In vitro investigations could also be carried out to study donkeys’ organ metabolism functions as done in horses [33].

Serum concentrations measured were low in the studied population compared to horses. The maximum mean serum concentration of the jennies was 0.28 ± 0.25 µg/mL following multiple doses at 18 h, whereas values from 0.32 ± 0.16 µg/mL (following a single dose) to 0.82 ± 0.13 µg/mL (following multiple doses) were reported in horses [9,11]. Interestingly, 2 of the 8 jennies presented two serum DXC peaks following the first administration (#1 and #8, Figure 2). Two peaks following oral DXC administration have been documented in some healthy horses [8,10], and in infected horses [34]. Some of these horses were fasted prior to DXC administration, therefore the interaction between DXC and food might not explain this phenomenon. It is hypothesized this can be from enterohepatic recirculation of DXC.

Despite a high V_z_/F, the measured concentrations of DXC in urine, synovial fluid, and endometrium were low, this could support the hypothesis of a lower V_d_ than V_z_/F. In the present study, the mean urine DXC concentration was lower than the mean serum DXC concentration, whereas it reached 200 to 300 times the mean serum C_max_ in healthy horses medicated with oral DXC at the same dosage regimen [9]. This suggests a lower renal clearance of DXC in donkeys compared to horses. However, higher metabolism, intestinal clearance, or V_d_ could also lower the concentration of DXC in urine. In the present study, the mean concentrations of DXC in synovial fluid were lower than in the serum, which suggests that accumulation of DXC in synovial fluid may not occur in the donkey, as it does in healthy horses [9,12]. In healthy horses receiving the same dosage regimen of DXC, the mean endometrial DXC concentration was 1.3 ± 0.36 µg/mg, which was 3.6 times the mean serum DXC concentration 3h after the last administration [9]. In the current study, because endometrium biopsies were not weighed, the endometrial concentration of DXC could not be reported. However, it seems unlikely that similar concentrations as to those in horses would be reached, because 3h after last administration, mean DXC measured in biopsies was only 0.13 ± 0.06 µg. Because calculated absorption (K_a_) and V_z_/F were high, but the measured concentrations of DXC in tissue and body fluids were low in the present study, it is hypothesized that the metabolism of DXC is different in donkeys than in horses. These results may be due to the higher metabolic rates and cellular water content in this species compared to horses [4]. This is consistent with previous findings of higher clearance, lower mean residential time and shorter half-life of most antibacterials in donkeys compared to horses [4].

The calculated absorption of DXC in donkeys in the present study was high (10.3 h^−1^), but bioavailability cannot be calculated with the data from this investigation. The determination of oral DXC bioavailability requires IV administration of DXC to compare the area under the curve of serum DXC concentration between oral and IV administration. Intravenous DXC administration results in cardiovascular toxicosis in horses and ponies, with potentially lethal outcomes [35], at dosages as low as 3 mg/kg [13]. At 10 mg/kg, intragastric DXC administration to non-fasted horses had a bioavailability of 17%. In top dressing pellet, oral ingestion of DXC had a bioavailability of 6% [13]. In another study, bioavailability following intragastric DXC administration at 20 mg/kg to fasted horses was estimated at 2.7% by allometric analysis [10]. However, an allometric analysis might not be a suitable technique to estimate the bioavailability of DXC in equids, as it extrapolates data from other species [36]. Therefore, the intestinal absorption of DXC in donkeys reported in the present study can be underestimated from concurrent high tissue distribution.

In the present study, DXC mixed in water was administered through a nasogastric tube, as is done in most pharmacokinetic studies in horses [9,10,11,12,13]. This is not a practical method of drug administration to all ill animals. The PK of DXC mixed in water and administered through oral dosing syringes has been investigated in horses [14]. Following a single administration of DXC mixed with water at 10 mg/kg and administered with oral syringes to healthy horses, the serum concentrations were comparable to those reported following intragastric administration [9]. This method of administration should therefore be investigated in donkeys.

To the authors’ knowledge, there is no report of the common bacterial isolates in donkeys and their MIC. In equine, bacterial pathogens are reported to be susceptible to DXC with an in vitro MIC lower than 0.25 μg/mL including all coagulase-positive, β-lactamase producing staphylococci, some *Rhodococcus equi*, *Streptococcus equi* sbsp. *equi*, *S. dysgalactiae* sbsp. *equisimilis*, some *S. zooepidemicus, Acinetobacter* spp., most *Pasteurella* spp., some *Actinobacillus equuli* and *Taylorella equigenitalis* [9,37]. *S. zooepidemicus* was the primary isolated pathogen in horses [38] and donkeys seemed highly susceptible to secondary infection of this opportunistic bacteria [39]. Doxycycline is described as a co-dependent bacteriostatic antibacterial [7], for which both time-dependent (T > MIC) and concentration-dependent (C_max_/MIC, AUC/MIC) indices have been reported predictive for clinical efficacy [40,41]. However, these pharmacokinetic/pharmacodynamic (PK/PD) indices are specific for combinations of pathogens and species. To the authors’ knowledge, no report of such indices has been published in the equine species. In addition, it is recognized that only unbound or free drug is biologically active [42,43]. However, the binding-unbinding protein phenomenon is a dynamic system; protein-bound drugs can dissociate when the protein-unbound drug is removed from the system, such as cell penetration. Therefore, controversies exist regarding the prediction of biological activity of measured unbound-fraction of the antibiotic [44]. To the authors’ knowledge, no PK/PD studies are reporting the prediction of biological activity of protein-unbound DXC in equine species. In the current study, all DXC (protein bound and unbound) was measured. A study reported DXC to be 82% protein-bound in equine plasma [10]. Therefore, in the current study, the reported values of DXC serum concentration may overestimate the concentration of biological active DXC. Overall, considering the MIC of the main isolated equine bacteria [9,37], and based on the present preliminary findings, oral DXC at 10 mg/kg q12 h seems unlikely to be a suitable dosage option for the treatment of infectious disease in donkeys. However, PK/PD integration model studies should investigate the optimal oral DXC dosages for donkeys for the treatment of specific bacterial isolates in this species [42,45].

Analysis of DXC was not performed by liquid chromatography-mass spectrometry (LC-MS) or HPLC because of unavailability at the host institution. While ELISA can be a less sensitive analytical method compared to LC-MS, several studies have shown there are no significant differences in sensitivity between LC and ELISA kits for some veterinary drugs such as flunixin, ractopamine, and chloramphenicol [46,47,48,49]. Studies in minor species are often financially limited, and some in-house laboratories may not be able to invest in additional equipment. Economical and less labor-intensive techniques such as ELISA could allow pharmacological studies to be carried out without the need to invest in more expensive equipment such as that required for LC-MS/HPLC. Therefore, further studies comparing the sensitivity of analytical methods such as ELISA, LC-MS and/or HPLC should be pursued.

In the current study, the serum concentrations were low, but no measured values were below the limit of quantification for the ELISA assay. This suggests the sensitivity could still be high for this method of analysis. When terminal samples are not below the limit of quantification, several pharmacokinetic parameters, such as terminal half-life, the V_d_, and mean residence time can be increased as an artifact of assay sensitivity [50]. Overall, validation of the ELISA kit for measuring DXC in donkey tissues in comparison to LC-MS is warranted for future studies planning to use this analytical technique.

## 5. Conclusions

This preliminary study reports pharmacokinetics parameters of DXC following oral administration in donkeys at 10 mg/kg q12 h and does not support this dosage regimen as a suitable treatment in donkeys until proven efficacious. This report can be used for designing future studies of DXC in donkeys. Authors recommend (1) validating the analytical method of the ELISA kit used, (2) sampling tissues and body fluids more frequently to use a more conventional pharmacological analysis (such as non-compartmental method), (3) sampling additional tissues to investigate the V_d_, and (4) investigating the route of elimination of DXC in donkeys, and the main pathogens affecting donkeys and their antibacterial susceptibilities.

## Figures and Tables

**Figure 1 animals-11-02047-f001:**
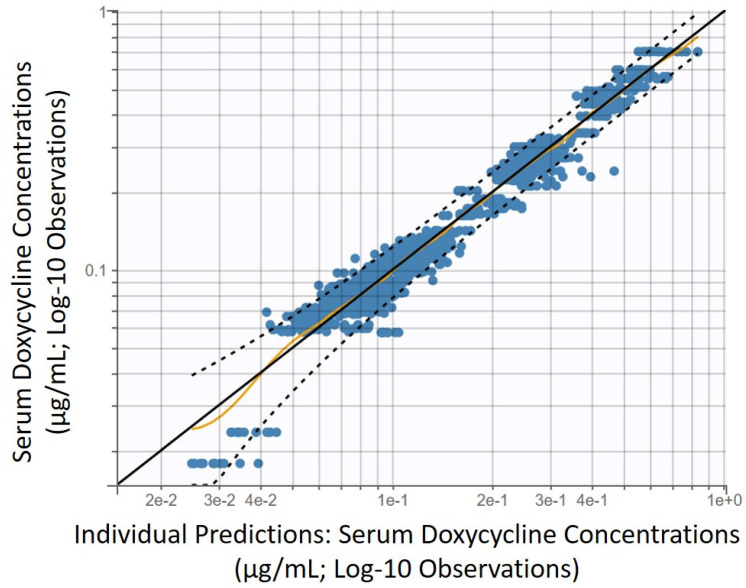
Predictive performances and robustness of fit of the final pharmacokinetic model of serum doxycycline concentration following administrations of 10 mg/kg doxycycline PO q12 h in 8 healthy jennies. Blue dots: observations; solid tan line, identity line; dotted black lines: 90% prediction interval.

**Figure 2 animals-11-02047-f002:**
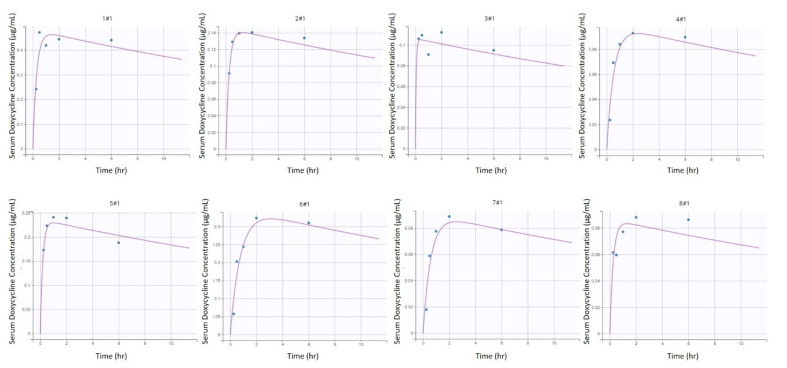
Individual predictions of initial doxycycline equivalent serum concentrations in donkeys from the final selected model (*n* = 8). Scatter plot of observed (blue dot) and predicted (dashed purple line) individual concentration vs. time after dosing following the first intragastric administration of doxycycline at 10 mg/kg in 8 healthy jennies.

**Figure 3 animals-11-02047-f003:**
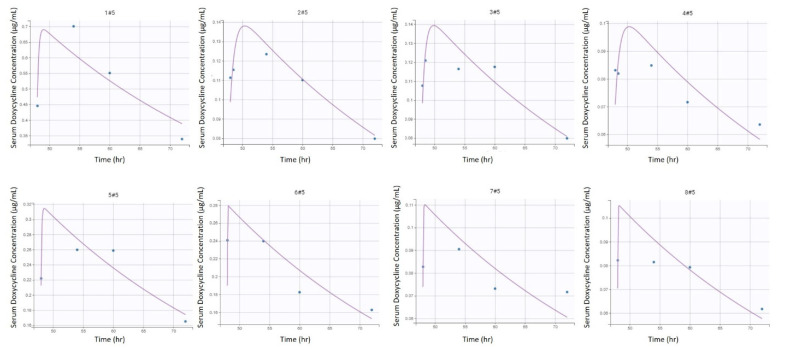
Individual predictions of final doxycycline equivalent serum concentrations in donkeys from the final selected model (*n* = 8). Scatter plot of observed (blue dot) and predicted (dashed purple line) individual concentration vs. time after dosing following the fifth intragastric administration of doxycycline at 10 mg/kg q12 h in 8 healthy jennies.

**Figure 4 animals-11-02047-f004:**
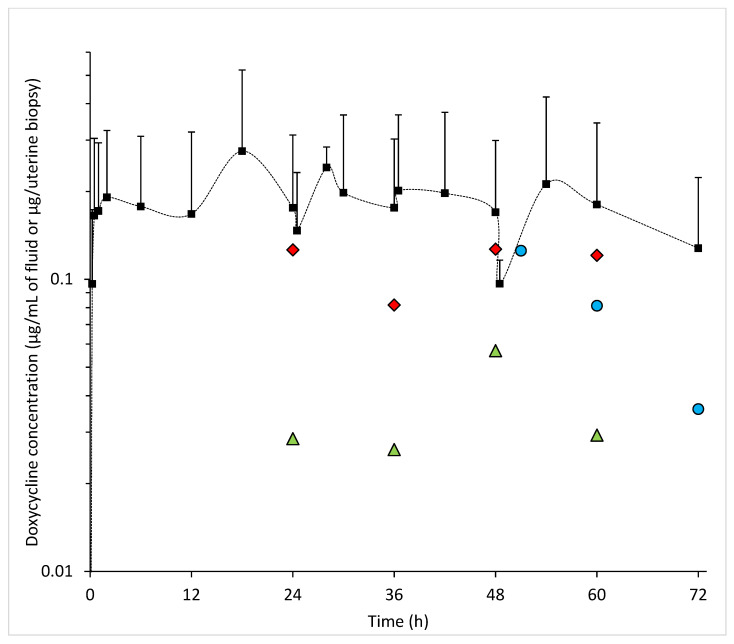
Mean + SD serum doxycycline concentration (square), and means of doxycycline concentrations in urine (red diamond), synovial fluid (green triangle) and endometrium biopsy (blue circle) vs. time during intragastric administration of doxycycline at 10 mg/kg q12 h from 0 to 48 h in 8 healthy jennies. The dash line connects the serum data point for ease of reading but does not represent continuum data.

**Figure 5 animals-11-02047-f005:**
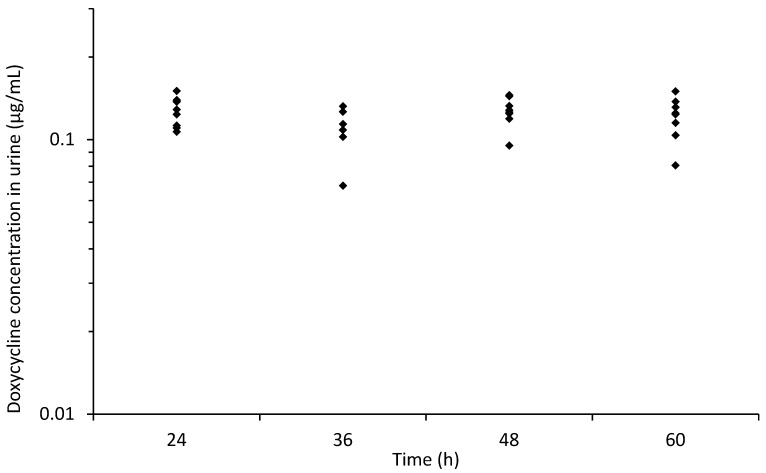
Individual concentration of doxycycline in urine vs. time during intragastric administration of doxycycline at 10 mg/kg q12 h from 0 to 48 h in 8 healthy jennies. Each solid diamond represents the value from one jenny at each time point.

**Figure 6 animals-11-02047-f006:**
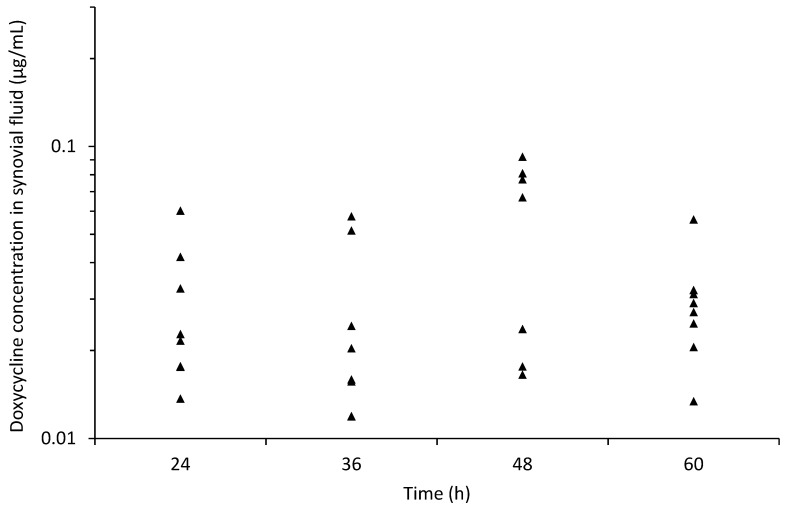
Individual concentration of doxycycline in synovial fluid vs. time during intragastric administration of doxycycline at 10 mg/kg q12 h from 0 to 48 h in 8 healthy jennies. Each solid triangle represents the value from one jenny at each time point.

**Figure 7 animals-11-02047-f007:**
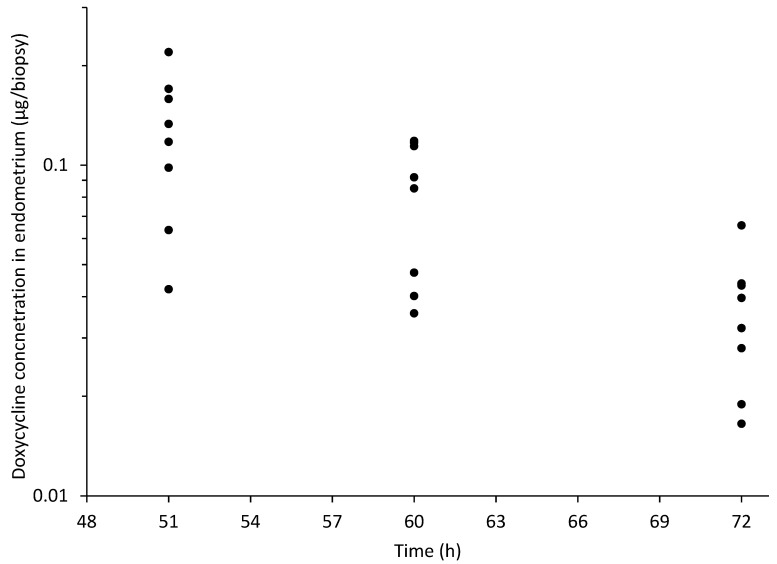
Individual concentration of doxycycline in endometrium biopsy vs. time following intragastric administration of doxycycline at 10 mg/kg q12 h from 0 to 48 h in 8 healthy jennies. Each solid circle represents the value from one jenny at each time point.

**Table 1 animals-11-02047-t001:** Estimated population model parameters for serum doxycycline concentration after intragastric administration of doxycycline at 10 mg/kg q12 h from 0 to 48 h in 8 healthy jennies.

Parameters	Unit	Point Estimate	Relative Standard Error %
K_a_ pop.	h^−1^	10.3	45.3
V_z_/F pop.	L/kg	108	24.4
K_el_ pop.	h^−1^	0.0253	9.82

K_a_, absorption rate constant first order; V_z_/F, apparent volume of distribution during terminal phase after non-intravenous administration; K_el_, elimination rate constant from the central compartment.

**Table 2 animals-11-02047-t002:** Individual for serum doxycycline concentration after intragastric administration of doxycycline at 10 mg/kg q12 h from 0 to 48 h in 8 healthy jennies.

Parameters	Minimum	Quartile Q1	Median	Quartile Q3	Maximum
K_a_	0.102	1.89	20.4	60	171
V_z_/F	20.9	62.5	115	204	311
K_el_	0.0253	0.0253	0.0253	0.0253	0.0253

K_a_, absorption rate constant first order; V_z_/F, apparent volume of distribution during terminal phase after non-intravenous administration; K_el_, elimination rate constant from the central compartment.

## Data Availability

Data is contained within the present article and Appendix A. Raw data collected presented in this study are available on request from the corresponding author.

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
