# Peer review of "Nonlinear Mixed-Effect Pharmacokinetic Modeling and Distribution of Doxycycline in Healthy Female Donkeys after Multiple Intragastric Dosing–Preliminary Investigation"

_animals, 2021, doi:10.3390/ani11072047_

Round 1

Reviewer 1 Report

The title adequately reflect the major finding of the study.

The abstract adequately summarize methodology, results, and significance of the study.

The introduction section falls within the topic of the study, however, the punctuation and English language should be improved. Moreover, I suggest to better emphasize the current popularity of donkey species and the  restrictions in the use of drugs in this species. At this regard, after “…increasing the demand for veterinary medical management [1].” (Line 51) I suggest to add: “In recent years, donkey farming gained popularity in several countries  where these equids are mainly reared for milk production (Veneziano V, et al., Vet J. (2011) 190:414–5). Thanks to its special properties, donkey milk is suitable for infants who cannot be breast-fed and people suffering from cow’smilk protein allergies (Bazzano M, et al., J Equine Vet Sci. (2019) 78:112–6). Donkeys reared for milk production needed to be continuously managed and monitored to maintain optimal general health conditions, avoiding nutritional deficiencies (Carroccio A, et al., Clin Exp Allergy. (2000) 30:1597–603). The restrictions in the use of drugs in food-producing animals has enforced the search for sustainable alternative approaches for treatment (Arfuso F, et al., Front. Vet. Sci. (2020) 7:556270).”

The section of Materials and Methods is clear for the reader and it meticulously describes the methods applied in the study. However, some missing information should be added.

Line 120: please add the manufacture and country of tubes used for blood collection.

Line 124, 129: please indicate the centrifugation speed and time applied on the samples.

Regarding data analysis, has a normality test been performed in order to assess the normal distribution od data?

Please indicate the results of correlation analysis (P value and Pearson’s r ) and, whether possible, of regression model (r2 value)

Results section as well as Discussion section is clear and well written. The findings obtained in the study were well discussed and justified with appropriate references.

The conclusion section should summarizes the results and the significance of the study. Please improve the sentence “In conclusion, this preliminary study reports pharmacokinetics parameters of DXC 393 following oral administration in donkeys at 10 mg/kg.” indicating that (as in abstract section) the oral doxycycline at the horse dosage should not be considered a suitable treatment in donkeys until proven efficacious.

Table and figure files are generally good and well represent the results. However, please better indicate the axis title (e.g. in Figure 2 please specify “doxycycline serum concentrations ug/mL”, and also for the other figures).

Authors should check and standardize the references in the list according to journal guidelines.

Reviewer 2 Report

This manuscript describes a pharmacokinetic (PK) study of doxycycline in female donkeys where serum, urine, synovial fluid and endometrium biopsy concentrations have been determined. The authors have adopted to use a NLME PK methodology to analyse the serum for the determination of PK parameters which are then compared to studies in the horse.

Major comments: The methods for the NMLE methodology have not been described fully and or are not clear as pointed out below.

Line 115: Correlation of parameters are described but no results shown. What was the correlation matrix for parameters as well as random effects? Were random effects correlated, partially correlated or not correlated in the final NMLE model?

As the study had information on body weight and i assume when the donkeys were fed why were they not included as co-variates in the model on fixed effects? 

What distribution was used for the random effects eg normal, log normal etc?

what weighting was used for the residual error of the model fit eg additive, proportional etc?

Did the authors think of doing a cross validation boot strap of the model ie removing data randomly and rebuilding the model and then compared to final original model?

Line 237: Table 1 displays population model parameters for the serum data. Are these population parameters the typical values (i.e fixed values) for the population? or are they the average of the post-hoc parameters values? Either way the typical values should be presented along with the post-hoc values for each donkey as there is only 8 animals.

I have major concerns with regard to the interpretation of the parameters from this study especially when compared to other horse studies. The authors have fitted the most parsimonious model to the serum data but that does not mean it is mechanistically correct. For instance, the rise in serum which has been fitted to an absorption phase may not be reflective of the true absorption rate as there may be rapid distribution occurring as well as absorption. This would mean that the absorption rate constant is a hybrid of initial distribution and absorption. The authors should comment on this in the manuscript. Furthermore, the Vd described by the authors is in fact a Vz/F i.e. the distribution in the terminal phase adjusted for bioavailability. If bioavailability varies because of the formulation and route of administration then Vz/F will also vary. The authors need to discuss this in the manuscript. 

Line 284: The authors suggest that the concentrations are low due to high clearance and low levels are associated with high Vz/F. This is a false conclusion for steady-state Vd as clearance and Vdss are independent of each other. Only if Vz/F has a small contribution to the area under the curve will clearance impact on the Vz/F. Authors should look to see what percentage of the AUC does the terminal phase contribute. This can be done with the model fit simulated data with non compartmental analysis. 

Half-life is a composite of clearance/F and Vz/F. As the half-life appears longer in this donkey study and the belief is that CL/F is higher, then the longer T1/2 must be related to a larger Vz/F. Authors should comment on this in the manuscript.

A higher metabolism in donkey compared to horses will lead to lower concentrations of drug in urine for the same renal clearance. This point needs to be explained in the manuscript as the authors don't give an explanation only that it needs to be investigated. 

line 307: the authors suggest that serum concentrations were low in this study compared to horses. 0.28+- 0.25 is arguably no different to 0.32 +-0.16 for a single dose. Can the authors make a stronger argument that there is a difference please. 

Minor comments: Line 245 and throughout the manuscript. The authors switch between the use of serum and plasma as though they are interchangeable. Please use the correct sample used throughout.

The authors on 2 occasions refer to doing future studies to show a difference between horse and donkey metabolism for the drug. I would recommend that the authors actually say what study eg investigations using horse and donkey hepatocytes kinetic studies. 

In several places the authors refer to an MIC of 0.25 ug/ml. Is the MIC from in vitro work or determined from serum/plasma. If the former then corrections need to be made for serum/plasma protein binding as only the free concentration can drive the pharmacological effect. Can the authors clarify MIC and if from in vitro corrections for serum binding need to be made when comparing serum concentrations to MIC.

Round 2

Reviewer 2 Report

The authors have made sufficient improvements to warrant publication - so I am happy with the revised manuscript.